# Proposal of *Bacillus altaicus* sp. nov. Isolated from Soil in the Altai Region, Russia

**DOI:** 10.3390/ijms26199517

**Published:** 2025-09-29

**Authors:** Anton E. Shikov, Maria N. Romanenko, Fedor M. Shmatov, Mikhail V. Belousov, Alexei Solovchenko, Olga Chivkunova, Grigoriy K. Savelev, Irina G. Kuznetsova, Denis S. Karlov, Anton A. Nizhnikov, Kirill S. Antonets

**Affiliations:** 1All-Russia Research Institute for Agricultural Microbiology, 196608 St. Petersburg, Russia; a.shikov@arriam.ru (A.E.S.); m.romanenko@arriam.ru (M.N.R.); a.nizhnikov@arriam.ru (A.A.N.); 2Faculty of Biology, St. Petersburg State University, 199034 St. Petersburg, Russia; 3Department of Bioengineering, Faculty of Biology, Moscow State University, 119234 Moscow, Russia; solovchenko@mail.bio.msu.ru (A.S.);

**Keywords:** Altai Republic, *Bacillus altaicus*, soil, BGC, genome assembly, comparative genomics, PANC-1, cytotoxicity, enterotoxins

## Abstract

The Altai Republic remains a geographic region with an uncovered microbial diversity hiding yet undescribed potential species. Here, we describe the strain al37.1^T^ from the Altai soil. It showed genomic similarity with the *Bacillus mycoides* strain DSM 2048^T^. However, the in silico DNA–DNA hybridization (DDH) was 61.6%, which satisfies the accepted threshold for delineating species. The isolate formed circular, smooth colonies, in contrast to the rhizoidal morphology typical of *B. mycoides*. The strain showed optimal growth under the following conditions: pH 6.5, NaCl concentration 0.5% *w*/*v*, and +30 °C. The major fraction of fatty acids was composed of C_16:0_ (34.77%), C_18:1_ (15.20%), C_14:0_ (9.06%), and C_18:0_ (7.88%), which were sufficiently lower in DSM 2048^T^ (C_16:0_–15.6%, C_14:0_–3.7%). In contrast to DSM 2048^T^, al37.1^T^ utilized glycerol, D-mannose, and D-galactose, while being unable to assimilate D-sorbitol, D-melibiose, and D-raffinose. The strain contains biosynthetic gene clusters (BGCs) associated with the production of fengycin, bacillibactin, petrobactin, and paeninodin, as well as loci coding for insecticidal factors, such as Spp1Aa, chitinases, Bmp1, and InhA1/InhA2. The comparative analysis with the 300 closest genomes demonstrated that these BGCs and Spp1Aa could be considered core for the whole group. Most of the strains, coupled with al37.1^T^, contained full *nheABC* and *hblABC* operons orchestrating the synthesis of enteric toxins. We observed a cytotoxic effect (≈19 and 22% reduction in viability) of the strain on the PANC-1 cell line. Given the unique morphological features and genome-derived data, we propose a new species, *B. altaicus*, represented by the type strain al37.1^T^.

## 1. Introduction

The Altai region represents a unique center of biodiversity of mountain plant and animal species in northern Asia, including endemics. Despite ongoing research efforts in this region, there is a huge gap in the characterization of microbiota occupying this heterogeneous zone. Several reports described bacilli isolated from this area, including *Bacillus anthracis* [1,2], *B. cereus* var Thuringiensis [3], and *B. pumilis* [4]. In general, the genus *Bacillus*, along with *Streptomyces*, is predominant in certain soils from rocks scattered in the Altai Republic [5]. Various soil dwellers in Altai represent potential biocontrol agents with antibacterial and fungicidal activities [5]. The whole region, due to distinct environmental gradients, is an eminent source of novel microorganisms [6,7,8]. The mentioned bacilliform bacteria occupying the diverse ecotopes from the Altai region belong to the *Bacillus cereus sensu lato* group with entangled taxonomy. On that account, comprehensive studies uniting both morphological and genomic approaches are required for a better resolution of such a complex phenomenon.

Our research contributes both to the identification of microbiota from Altai and the development of *B. cereus s. l.* classification. Here, we isolated the strain al37.1^T^, demonstrating a considerable similarity with *B. mycoides*, a spore-forming, Gram-positive bacterium, which is widely distributed in soil and rhizosphere [9]. *B. mycoides* remains poorly studied, unlike its within-genus counterparts, such as *B. cereus* and *B. anthracis* [9]. Nonetheless, some *B. mycoides* isolates have demonstrated beneficial plant growth-promoting (PGP) and biocontrol properties in crops such as sunflower [10], cucumber [11], and sugar beet [12]. In this context, studying *B. mycoides* genomes and the related species can help uncover genomic determinants orchestrating the biosynthesis of compounds potentially responsible for agriculturally promising features.

We primarily aimed to dissect the taxonomic position of the isolate al37.1^T^ from the Altai soil. It exhibited a considerable dissimilarity with *B. mycoides* according to genomic comparative metrics, morphological features, and biochemical characteristics. Given the observations obtained, we propose a novel species, *Bacillus altaicus* sp. nov., uniting at least three known strains, including ours, based on genomic data. By employing a large-scale pangenomic analysis of the closest reference genomes from the *B. mycoides* group, we identified metabolic clusters and operons responsible for the biosynthesis of enterotoxins. We revealed a moderate yet significant cytotoxic effect of al37.1^T^, which shows the necessity to verify carefully the safety of the strains close to *B. mycoides* when applying them in agriculture.

## 2. Results

### 2.1. General Description and Morphological Characteristics of the Bacillus sp. al37.1 Strain

#### 2.1.1. Initial 16S rRNA Sequence Analysis Suggests That the Strain Is Close to *Bacillus mycoides*

A total of 32 strains were isolated from soil samples collected in the Altai region, including strain al37.1, inhabiting the meadow soil and described in this article. The strain was deposited in the joint Russian Collection of Agricultural Microorganisms (RCAM) at the All-Russia Research Institute for Agricultural Microbiology in Saint Petersburg (https://rcam.arriam.ru/, accessed on 17 September 2024) in December 2023 under the registration number RCAM06651. The analysis of 16S rRNA loci obtained with Sanger sequencing demonstrated that the strain most probably belonged to the *B. mycoides* species (99.9% similarity). On that account, further experimental and genomic studies were carried out mainly in comparison with this species.

#### 2.1.2. Assessing the Optimal Growth Conditions of the Strain

The al37.1 isolate grew within a pH range of 6.0–9.0, with maximum NaCl tolerance (up to 5%, *w*/*v*) observed at pH 8.0. At pH 7.0, growth was supported up to 3% NaCl, whereas at pH 6.0 and 9.0 it was limited to 1% NaCl. The strain grew at temperatures between +20 and +35 °C, with optimal growth at +30 °C. Based on these results, subsequent experiments were conducted in standard 2YT medium [13] (pH 6.5, NaCl concentration 0.5% *w*/*v*).

#### 2.1.3. Morphological and Physiological Properties of the Isolate Distinguish It from Known Species

The colony morphology of al37.1 differed from the typical rhizoid shape, being a main characteristic of the putative candidate species, *B. mycoides* [14]. Under optimal growth conditions, colonies were circular, 2–4 mm in diameter, with a rough surface, convex elevation, smooth margins, and matte yellowish-white pigmentation (Figure 1A). They had a finely granular texture and were easily removed from the agar surface.

Gram staining showed positive results for al37.1 (Figure 1B). Light microscopy showed that cells were rod-shaped and often arranged in chains. Oval endospores were formed and located centrally within the cells (Figure 1C).

Further observations using transmission electron microscopy provided additional detail on cellular structure. Rod-shaped cells measured 1.5–4.3 μm in length and 0.6–1.8 μm in width and possessed filamentous structures resembling polar flagella (Figure 2A–C). Light microscopic examination revealed a smooth movement of all bacterial cells in a single direction, consistent with a convection current, indicating that the observed filaments are likely non-functional for motility. Endospores with a surrounding endosporium were also visualized within the cells (Figure 2D). The spores measured 1.1–2.2 µm in length and 0.6–1.1 µm in width.

### 2.2. Genomic Analysis Suggests That al37.1^T^ Represents a Novel Species, Bacillus altaicus

To employ comparative genomic analysis and inspect the taxonomy of the strain on a genome level, we performed the whole-genome sequencing of the strain al37.1. Having processed and inspected the raw sequencing data from using Illumina and Nanopore platforms, we obtained a high-quality complete-level genome consisting of two circular contigs. The genome was 5,397,466 bp in length, and the N50 reached 5,117,040 bp. The assembly was deposited in the NCBI RefSeq database under the accession number GCF_042136375.1. The coverage depth reached 495 reads per bp. Genome completeness constituted 98.02%, with only 0.85% of contamination. The largest 16S rRNA locus was 100% identical to the Sanger-based sequence. The complete single-copy orthologs relative to the bacillales_odb10 BUSCO [15] database reached 98.9%. The genome contained 5306 CDS (coding sequences). Therefore, according to the genomic properties, the annotated genome assembly was of high quality to fully represent the studied strain.

Having obtained 16S rRNA-based evidence on the strain to resemble *B. mycoides*, we compared it with the type strain of the species, DSM 2048^T^, and revealed that the in silico DDH estimate reached 61.6% which is below the species delineation threshold (Table 1). However, the closest genomes from the NCBI RefSeq [16] database represented those with ambiguous taxonomic categories. Therefore, we prepared a genomic dataset by picking the related assemblies according to ANI (Appendix A). We selected all representatives labeled as *B. mycoides*, coupled with other species, if the ANI values were higher.

With a dataset of 300 genomes and the outgroup *B. subtilis* strain 168, we prepared a pangenome-based reference phylogeny (Figure 3). In the reconstructed pangenome, only 5.7% represented core genes, and the pangenome was open. According to the collected metadata, the dataset included 8 defined species, 155 of which belonged to *B. mycides* (Figure 3A; Appendix A). Most of the strains were isolated in the USA (62), Poland (37), and Germany (37). In general, the genomes corresponded to soil-dwelling (115), rhizospheric (50), and those isolated from food sources (42). Except for the groups of highly close isolates, the strains with distinct origins were dispersed along the tree (Figure 3A), while the erroneous self-titled species attributions were vivid since multiple strains attributed to *B. mycides* were separated in clades of distinct taxonomic composition (Figure 3A).

The assemblies were highly homogenous in GC content, reaching 35.3 mol% on average (34.8–35.7%; Figure 3B), while only the *B. subtilis* strain 168 varied substantially (43.5 mol%). In general, all the strains except the outgroup were low-recombining, with an r/m (recombination-to-mutation) rate spanning from 0.39 to 1.85 (0.91 on average; Figure 3B). The isolate al37.1^T^ fell into a single compact clade with 42 strains close to the largest subtree encompassing the representatives of the *B. mycides* species meeting the proposed genomic criteria of 95% ANI and 70% DDH (Figure 3C; Appendix A). The whole clade was characterized by a drop in the mean DDH to 59.1% relative to the type strain DSM 2048^T^ (Figure 3C). The ANI values, albeit being relatively high (94.7%), fell below the species delineation threshold except for the three strains, including ours, in a small subclade (Figure 3C). Only two strains (FSL R9-9410 and BPN57/2) from the compact clade fell into a single species with al37.1^T^ according to the DDH estimates (Figure 3D).

Next, we analyzed sequences of the established taxonomic genome feature, 16S rRNA-encoding loci. To circumvent the presence of fragmented contigs, we used local similarity between sequences. A high similarity with an average of 99.2% was detected. Aside from the *B. subtilis* 168 (94.2% on average, 93.9% with *B. mycides* DSM 2048^T^). The ML phylogeny reconstructed on 16S rRNA sequences and the dendrograms differed from the reference ones (54.7–63.1% topological similarity), indicating that the taxonomic assignment procedure should consider complex factors.

We then applied the single- and multi-loci-based phylogeny reconstruction, considering the *gyrB* locus and *B. cereus* MLST scheme (Multilocus Sequence Typing). Importantly, both trees retained the isolated compact clade with the strain al37.1, corroborating its distinct genomic nature (Appendix A). Unlike the 16S rRNA gene, the similarity of the *gyrB* locus from al37.1 relative to *B. mycides* DSM 2048^T^ was considerably lower (97.77%).

To classify the allele composition of the genes in the MLST scheme, we applied the mlst v2.23.0 software and collected the resulting allelic profiles. In contrast to type strains of the known species, the al37.1 strain lacked any known complex (Appendix A). The characteristics of individual genes in the conventional *B. mycoides* clade were homogeneous in contrast to the compact clade with the strain al37.1 (Appendix A). Sequence similarity with the DSM 2048^T^ hovered around 95.6–98.9% (Appendix A). Notably, the phylogeny based on *gyrB* was more topologically similar to the pangenome-wise tree (82%) in comparison with the phylogeny reconstructed on the components from the MLST scheme (72%).

Since the taxonomy of the *B. cereus s. l.* is quite intricate, we applied the Btyper3 v3.4.0 [17] tool to disentangle genomospecies in our dataset. A total of four species, namely, *B. mycoides*, *B. toyonensis*, *B. paramycoides*, and *B. luti*, belonging to four PanC groups, were reported (Figure 3E). Both *panC* typing and the joint Btyper3 classification failed to distinguish the *B. mycoides* species on the ML tree unambiguously (Figure 3E), whereas three other species were clade-restricted. According to these findings, the currently accepted *mycoides* group is a polyphyletic unit.

The whole clade attributed to the al37.1^T^ strain was classified as *B. mycoides*. The genomic characteristics of the type strains, whose genomes were present in the dataset (Table 1), however, illustrated that the al37.1^T^ strain represented a considerably close but separate species from *B. mycoides* since the DDH with DSM 2048^T^ reached 61.6%, gyrB sequences were different, and the MLST complex for the strain was unknown. Moreover, given the above-mentioned DDH distributions in the closest clade, we might conclude that there are two relative species, one of which includes the al37.1^T^ strain, termed *Bacillus altaicus* sp. nov. Since genomic similarities according to ANI estimates were strikingly high, we further analyzed the chemotaxonomic profile and physiological characteristics of the al37.1^T^ strain to prove its distinguished taxonomic category.

### 2.3. Bacillus altaicus sp. nov. Exhibits a Unique Composition of Fatty Acids and Biochemical Capabilities

#### 2.3.1. Biochemical Characteristics

Having shown a noticeable, albeit moderate, genomic evidence suggesting that the *B. altaicus* al37.1^T^ isolate is a novel species, we used a well-established workflow to reveal both physiological differences and chemical composition of the membranes.

The Biolog GEN III analysis revealed that *B. altaicus* al37.1^T^ utilized 50 carbon sources, including various carbohydrates, amino acids, peptides, and organic acids (Table 2). In contrast, the strain was unable to metabolize dextrin, gentiobiose, sucrose, stachyose, D-raffinose, α-D-lactose, D-melibiose, N-acetylneuraminic acid, D-sorbitol, D-mannitol, D-aspartic acid, gelatin, L-pyroglutamic acid, pectin, D-galacturonic acid, mucic acid, citric acid, D-malic acid, Tween 40, γ-aminobutyric acid, and acetoacetic acid. It also showed tolerance to 1% sodium lactate, guanidine hydrochloride, nalidixic acid, potassium tellurite, aztreonam, and sodium butyrate. These results indicate that *B. altaicus* al37.1^T^ possesses broad metabolic capacities and the ability to combat multiple chemical stressors.

When compared with its two closely related type strains based on DDH analysis, *B. mycoides* DSM 2048^T^ and *B. toyonensis* BCT-7112^T^, *B. altaicus* al37.1^T^ exhibited several distinct metabolic traits (Table 2). Unlike both type strains, it was able to utilize glycerol, D-mannose, and D-galactose. Both *B. altaicus* and *B. mycoides* metabolized D-arabitol, L-rhamnose, and D-fucose, which were not utilized by *B. toyonensis. B. altaicus* and *B. toyonensis* were unable to utilize D-sorbitol, D-melibiose, or D-raffinose, which were metabolized by *B. mycoides. B. altaicus* showed no utilization of sucrose, whereas both *B. toyonensis* and *B. mycoides* did. Overall, these differences highlight the unique combination of carbon source utilization patterns in *B. altaicus* al37.1^T^, distinguishing it from both phylogenetically close species.

#### 2.3.2. Chemotaxonomic Profile

After analyzing the physiological and biochemical signatures of the strain, we proceeded with identifying its marker fatty acid composition. The cellular fatty acid profile of *B. altaicus* al37.1^T^ was dominated by C16:0 (34.77%), C18:1 (15.20%), C14:0 (9.06%), C18:0 (7.88%), and C12:0 (4.26%). Minor components included C16:1 (4.38%), C14:1 (4.68%), C18:2 (4.39%), C13:0 (2.61%), C16:2 (5.52%), and other fatty acids present in amounts below 2%. Compared to *B. mycoides* DSM 2048^T^, al37.1^T^ contained higher proportions of C16:0 (34.77% vs. 15.6%), C14:0 (9.06% vs. 3.7%), and C18:0 (7.88% vs. 1.6%), while C12:0 was also more abundant in al37.1^T^ (4.26% vs. 2.7%). In addition, *B. altaicus* al37.1^T^ contained substantially higher proportions of the shared saturated fatty acids than *B. toyonensis* BCT-7112^T^, with C16:0 accounting for 34.77% (vs. 5.6%) and C14:0 for 9.06% (vs. 3.2%) (Table 3).

Given the evidence obtained, metabolic activities and the composition of fatty acids jointly corroborate that *B. altaicus* sp. nov. al37.1^T^ is a distinct taxonomic unit and should be treated as a separate species.

### 2.4. Comparative Genomic Analysis Reveals That B. altaicus and Its Closest Relatives Within the B. mycoides Group Are Enriched with Functional Loci

#### 2.4.1. *B. altaicus* and *B. mycoides* Group Species Hold Insecticidal Potential

We proceeded with a comparative analysis of the functional loci and predicted activity of the *B. altaicus* sp. nov. al37.1^T^ strain, coupled with its distant and close counterparts, starting with its virulence potential (Figure 4A; Appendix A). All the strains possessed at least one putatively insecticidal gene (Appendix A), with a maximum of 19 (JAS635 strain). The al37.1^T^ strain contained 7 loci, including those encoding the Tpp80Ab (31.6% identity) and Spp1Aa (79.7%) insecticidal toxins as well as Chitinases C and B, and proteases Bmp1 and InhA1/InhA2 (Figure 4A; Appendix A).

The outgroup *B. subtilis* 168 harbored only one gene encoding the toxin App6Ba (34.1%), while the remaining references and the strain al37.1^T^ were predictably able to produce InhA1/InhA2 (Appendix A; Appendix A). Other frequent toxins included Bmp1 (293 hits), Spp1Aa (282), and ChitinaseC (192). There were no obvious associations between gene presence and taxonomic groups, except for the ChitinaseC, predominantly found in the conventional *B. mycoides* species (Appendix A).

According to the predicted insecticidal activities taken from the BPPRC database [21], a wide range of affected hosts was found in the *B.* spp biovar Thuringiensis scattered along phylogenetic leaves (Appendix A). At the same time, since Spp1Aa has exerted the toxic effect on the order of Blattoidea [22], coupled with the abundance of the metalloproteases, one could expect that representatives of the *B. mycoides* group could likely exhibit insecticidal activities in general (Figure 4B).

**Figure 4 ijms-26-09517-f004:**
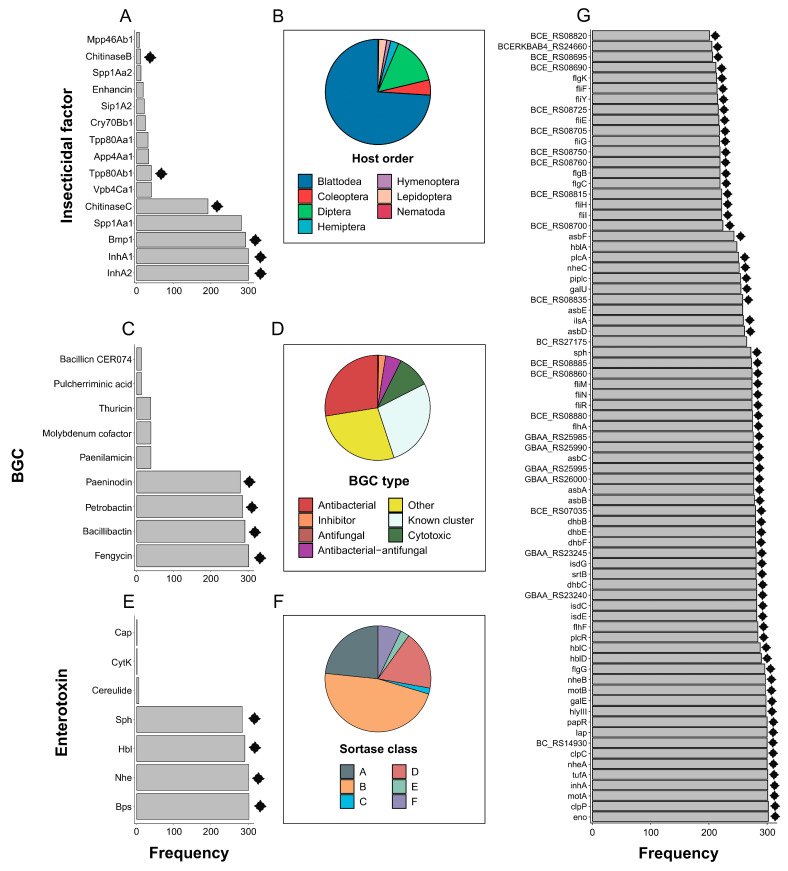
Summary of functionally important loci and predicted biological activities of the strains belonging to *B. altaicus*, *B. mycoides*, and close species. (**A**) Total frequency of insecticidal factors identified in strains (**B**). The summary of predicted insecticidal activities based on the data from the BPPRC specificity [21] database. (**C**) The spectrum of BGCs found in the analyzed genomes. (**D**). Classification of all BGCs detected in the dataset according to the biological activity predicted by the DeepBGC v0.1.30 [23] software. (**E**) The number of enterotoxin-related operons revealed in the genomic dataset with the Btyper3 v3.4.0 [17] utility. (**F**) The overall proportion of sortases attributed to the classes with the SortPred tool. (**G**) The frequency of virulence determinants representing homologs of the known virulence factors from the Virulence Factor Database (VFDB) [24] resource. The frequency corresponds to the number of genomes in which the loci were found. In plots (**A**,**B**), only hits revealed in at least 10 genomes are displayed. Similarly, plot (**G**) describes virulence factors found in more than 200 strains. Black diamonds mark the loci presented in the genome of the *B. altaicus* strain al37.1^T^. For the clade-wise representation of the respective data, see Appendix A. Detailed description of the factors is available in Appendix A.

#### 2.4.2. Certain BGCs Represent the Metabolic Core of the *B. mycoides* Group

Next, we described the metabolic properties of the *B. mycoides* operational group by collecting the data on genome-specific BGCs. A total of 5729 regions corresponding to known and predicted clusters were identified (Figure 4C; Appendix A). The most frequent chemical classes were BGCs synthesizing saccharides (945), unspecified ribosomally synthesised and post-translationally modified peptide products (RiPP-like) (694), and non-ribosomal peptide synthetase (NRPS) (396) products. The isolates harbored from 9 to 31 clusters, with a mean of 19 BGCs, and the strain al37.1^T^ contained 14 BGCs in the genome (Appendix A).

The strain al37.1^T^ was associated with four known BGCs, such as those responsible for the production of fengycin (40% similarity), bacillibactin (85.7%), petrobactin (100%), and paeninodin (80%). These clusters were the most frequent among the close reference strains, reaching 294, 284, 278, and 272 predicted BGCs in the analyzed genomes, respectively (Figure 4C). To improve functional predictions, we employed the prediction of the activity by the DeepBGC v0.1.30 [23] utility. The distribution of activities was uniform along all the analyzed strains (Appendix A). A total of 2325 BGCs putatively corresponded to bactericidal moieties, indicating the potency of the *B. mycoides* group as antibacterial agents (Figure 4D; Appendix A).

The presence of known BGCs illustrated that the four BGCs mentioned above represent a certain metabolic core of the *B. mycoides* group. A notable exception included *B. paramycoides* species, not capable of producing petrobactin (Appendix A). In general, we could propose that metabolic properties determined by the genomic landscape are considered genomic properties of the species group.

### 2.5. Genomic Inferences and Experimental Assays Demonstrate the Cytotoxic Properties of B. altaicus

#### 2.5.1. Representatives of the *B. mycoides* Group Are Enriched with Enterotoxins and Other Virulence Factors

Having analyzed metabolic and insecticidal properties, we next proceeded to study the biosafety of the selected strains. According to Btyper3 v3.4.0 results, our strain contained one gene (*bpsE*) from the capsule Bps antigen synthesis and complete operons responsible for the biosynthesis of enterotoxins Nhe and Hbl, as well as sphingomyelinase (Figure 4E; Appendix A). Full Nhe-producing gene sets were found in all *B. mycoides* close species, while Hbl biosynthesis clusters were identified in 292 assemblies (Appendix A).

We then mined for genes coding for sortases, cysteine transpeptidases serving as virulence determinants (Appendix A). The sortase classes were relatively homogenous, with 47% of hits representing the A class (Figure 4F). On average, the isolates contained 136 sortase-encoding loci ranging from 95 to 179, with 138 identified in *B. altaicus* sp. nov. al37.1^T^ and *B. subtilis* 168 (Appendix A). According to the homologs of the known virulence determinants from VFDB [24], the close genomes possessed from 38 to 105 factors (Appendix A), while the *B. subtilis* 168 strain harbored 18 virulence genes only. The strain al37.1^T^ possessed 86 virulence determinants (Figure 4G; Appendix A). The most abundant genes included *clpP* (302), *eno* (302), *motA* (301), *tufA* (301), and *clpC* (300). Similarly to insecticidal factors, apart from the outgroup, only sporadic small clade-scale taxonomic associations were found, such as the absence of certain components of the flagellin biosynthesis in a subclade of conventional *B. mycoides* and *B. paramycoides* (Appendix A).

In summary, the abundance of virulence determinants found in the *B. mycoides* group, in general, does not mirror the within-group phylogenetic differences and, contrarily, reflects either individual adaptation of the whole population properties. Virtually all the representatives of the dataset are enriched with similar virulence determinants, raising safety issues. To verify whether these genomic traces of potentially hazardous properties truly indicate the capability of exerting a deleterious effect on mammals, we employed a cytotoxicity assay.

#### 2.5.2. Cytotoxic Activity of *B. altaicus* al37.1^T^ Against PANC-1 Cell Line

The cytotoxicity effect of *B. altaicus* al37.1^T^ culture supernatants against the human pancreatic cancer cell line PANC-1 was examined using the MTT assay. After 24 h of incubation, filtrates obtained from distinct stages of cultivation exhibited variable inhibitory effects depending on both culture age and dilution (Figure 5).

At the highest concentration tested (1:10 dilution), the 7 h and 29 h culture filtrates significantly reduced cell viability by ~19% and ~22%, respectively, whereas after 16 h of incubation, no significant effect was observed. The toxicity displayed a visible dose dependency, being most pronounced at 1:10 with *p* < 0.0001, significant at 1:100 with *p* < 0.001, and substantially reduced at higher dilutions, being not significant at 1:1000.

Thus, the cytotoxic potential appears restricted to early and late growth phases, suggesting stage-specific production of active metabolites and demonstrating that genomic inferences marking virulence determinants are closely related to the observed cytotoxicity.

## 3. Discussion

In this study, we conducted a comprehensive analysis of the al37.1^T^ strain and assigned it to a novel species. The strain belongs to the *B. cereus sensu* lato group. This, strictly speaking, non-taxonomic unit represents the most intricate pool of species within the genus. Initially, it was composed of three species, *B. cereus*, *B. anthracis*, and *B. thuringiensis* [25]. With the advent of the next-generation sequencing technique and whole-genome-based comparisons, the number of species has continuously grown from 6 to up to 37 putative species [26]. To overcome such ambiguities, the pair-wise ANI (Average Nucleotide Identity) estimates forming the isolated connected components within the respective graph were proposed to describe genomospecies [27]. To date, *Bacillus cereus s. l.* encompassed 14 genomospecies; however, the reliance on a single criterion to distinguish bacterial species should be treated with caution [28]. Therefore, we considered multiple genomic traits, starting from the phylogenetic analysis.

The pangenome-wise tree demonstrated that our strain al37.1^T^ forms a sister clade with *B. mycoides* encompassing two expected species, one of which is *B. altaicus* sp. nov. proposed here. According to the ANI criteria, the similarity with the type *B. mycoides* strain exceeded 95%. However, the estimates were highly close to the species delineation threshold (Figure 3C,D). Such issues could lead to inconsistencies in separating species when distinct ANI calculation methods are applied and should be treated with caution accordingly [28]. The pairwise comparisons of 16S rRNA sequence similarity in the whole dataset illustrated high identity exceeding 99% corroborating that using this common criterion fails to distinguish *B. cereus s.l.* species [29]. On this account, we preferred DDH as a gold standard in bacterial taxonomy [30,31]. According to the well-established 70% DDH threshold, *B. altaicus* sp. nov. al37.1^T^ presented a novel species (Figure 3C,D). The distinct taxonomic attribution of *B. altaicus* was confirmed by *gyrB*-based phylogeny (Appendix A) and the unknown MLST profile (Appendix A). Higher topological similarity of the *gyrB* phylogeny with the pangenome-wide inference suggests its suitability for rapid typing of *B. mycoides*, which confirms the established views [29]. However, visible discrepancies in species distribution (Appendix A) and topological incongruence with the reference tree illustrate that establishing novel species requires a multi-factor description.

To prove the genomic inferences, we analyzed the morphology of the isolate, supporting its distinct taxonomic position. Although *B. mycoides* is generally defined by rhizoidal colony morphology [14], al37.1^T^ formed smooth, circular colonies with yellowish-white pigmentation. Rare reports [32] of atypical *B. mycoides* with similar non-rhizoid phenotype indicate that morphology alone cannot resolve separate species within this group. However, the unique morphology of al37.1^T^, coupled with genomic evidence, supports that *B. altaicus* represents a distinct taxonomic category. According to biochemical and physiological analyses, the strain can metabolize a wider array of carbon sources compared to its closest relatives, including substrates such as glycerol, D-mannose, and D-galactose. Another phenotypic characteristic, the chemotaxonomic profile, revealed that the al37.1^T^ strain exhibits higher proportions of saturated fatty acids, especially C16:0 and C14:0, compared to *B. mycoides* DSM 2048^T^ and *B. toyonensis* BCT-7112^T^.

The identified morphological and physiological features of the *B. altaicus* species probably reflect the ecological niche orchestrating the emergence of the species. The metabolic versatility mirrors the adaptive potential to thrive in heterogeneous soil environments of the Altai Republic with uneven nutrient availability and abiotic stresses. Modifications in fatty acid composition are known to influence membrane rigidity and stress tolerance, implying that this profile mirrors the adaptation to the extreme continental climate of the Altai region [33]. Previous studies demonstrated that attuning fatty acid composition orchestrates the adaptive potential of bacteria occupying harsh terrestrial niches [34,35,36,37]. While the gradients in organic compounds and mineral content on the Altai soil are expected to influence the *B. altaicus*, the exact relationships between soil composition and microbial diversity in Altai is not well understood. Current studies are focused primarily on specific niches, such as soda lakes, lignite coal seams, and bacterial mats [38,39,40,41]. However, several works revealed associations between microbial composition and the availability of nutrients in soil samples from kudurite rocks [5] and soils with distinct water content in the Altai region [42]. A more visible impact of altitudinal and horizontal gradients was studied on plants. For different plant species, gradients of nutrients, temperature, water availability, and mineral composition of soils affected the genotype, morphology, carbon content, and spectrum of synthesized metabolites [43,44,45,46]. All this evidence demonstrates that soil conditions indeed shape phenotypical features leading to speciation. Thus, further comparative research regarding a detailed description of the soil environment occupied by *B. altaicus* and *B. mycoides* is needed.

Since we obtained a high-quality genome assembly of the al37.1^T^ strain and showed its close position to *B. mycoides*, we summarized the predicted metabolic and virulent potential of the whole species group. *B. altaicus* sp. nov. al37.1^T^ contained genes encoding the Tpp80Ab and Spp1Aa toxins. Although Tpp80Ab remains unstudied, its closest relative, Tpp80Aa, exhibited pronounced activity against Diptera [47,48]. Spp1Aa, originally known as sphaericolysin, in turn, exerted the toxic effect on Blattodea and Lepidoptera in insects, namely, *Blattella germanica* and *Spodoptera litura*, respectively [22]. We demonstrated that Spp1Aa is the most widespread insecticidal factor in the *B. mycoides* species complex (Figure 4A). Given the concomitant high frequency of loci coding for InhA1/InhA2 metalloproteases, which cause histopathological effects on insects [49], Bmp1 with pronounced nematocidal activity [50,51], and chitinases with varied and target-specific activities [52], we could expect the non-Thuringiensis isolates from the *B. mycoides* group to represent prospective pesticidal agents.

Apart from insecticidal factors, the *B. altaicus* sp. nov. al37.1^T^ strain harbored 14 BGCs in total, four of which were responsible for the synthesis of known metabolites with broad biological activities. Fengycin, a cyclic lipopeptide, is recognized for its potent fungicidal and bactericidal properties against *Botryosphaeria*, *Xanthomonas*, *Pseudomonas*, and other phytopathogens [53,54]. Bacillibactin is a well-known triscatecholate siderophore [55], exhibiting strong bactericidal, fungicidal [56,57,58], and growth-promoting [55] properties. Petrobactin, another catecholate siderophore initially perceived as a *B. anthracis* virulence factor [59] and later found to be widespread around *Bacillus* species [60], due to its iron-chelating abilities, is advantageous for the plants by sequestering iron essential for plant pathogens [61]. Finally, paeninodin is a lasso peptide that belongs to the class of RiPPs (Ribosomally Synthesized and Posttranslationally Modified Peptides) [62]. While it does not inhibit bacterial growth per se, it was detected in bioactive strains of *Paenibacillus* associated with plant growth promotion [62]. This observation implies that paeninodin might enhance the overall antimicrobial activity when acting jointly with other bactericidal or fungicidal compounds. Such a combination of four BGCs is conserved among the *B. mycoides* group genomes (Appendix A). We previously reported the same observation on a smaller scale [63], which was preserved here when extending the genomic dataset.

While the *B. mycoides* group species certainly show pesticidal and bactericidal potential, the genomic survey raised safety issues as well. Compared to the *B. subtilis* outgroup, all the studied genomes, including *B. altaicus* sp. nov. al37.1^T^, were rich in genes associated with well-established virulence factors (Figure 4G,H), such as sortases contributing to bacterial pathogenesis [64,65] or Clp ATPases modulating the virulence in human pathogens [66]. Special attention should be paid to the enteric toxin-producing loci, namely, *nheABC* and *hblABC* operons detected in our strain and the vast majority of the isolates as well (Appendix A). The synthesis of the respective non-hemolytic enterotoxin and hemolytic enterotoxin hemolysin BL is associated with the infection outbreaks caused by *B. cereus s.l.* strains [67,68]. Since *B. mycoides* strains are frequently found in food sources [69,70] and are applied as biocontrol agents [12], which, by current reckonings, occasionally contain isolates showing pathogen-like activities in vitro [71], we should take into account possible detrimental qualities of the tested isolates. Nevertheless, the presence of the operons per se does not imply the synthesis of these toxins [72]; thus, both laboratory experiments and genomic surveys are needed when implementing the *B. mycoides* group representatives in bioformulations to ensure safe usage.

To confirm if the predicted virulence potential reflects biological activities, we assessed the cytotoxic effect of *B. altaicus* sp. nov. al37.1^T^ against the human PANC-1 cells (Figure 5). Intriguingly, we observed a dual response, with significant cytotoxicity (~19–22%) being detected at mid-exponential (7 h) and stationary (29 h) phases with no detectable effect in the transit phase (16 h). The transient disappearance of cytotoxic activity at 16 h could reflect CodY-mediated regulation in response to intracellular metabolite levels. CodY is triggered by binding GTP and branched-chain amino acids [73]. We propose that since in the mid-exponential growth (7 h) these metabolites are extensively used for the bacterial growth, a significant activation of CodY-mediated regulatory pathways initiates during the transition phase (16 h), accompanied by the accumulation of GTP and BCAAs. The signaling cascades could downregulate the expression of virulence genes. In addition, extracellular proteases secreted during this stage possibly contribute to the loss of cytotoxic activity by degrading proteinaceous toxins [74]. Once cells enter the stationary phase (29 h), nutrient depletion reduces CodY activity, while alternative regulatory mechanisms, such as the PlcR–PapR quorum-sensing system [75], are activated, eventually triggering the production of cytotoxic compounds. Given the presence of PlcR–PapR components, coupled with the *nheABC* and *hblABC* operons in the genome of strain al37.1^T^, the observed effects are most likely attributable to the activity of the Nhe and Hbl enterotoxins, in agreement with the genomic predictions. Taken together, the phase-dependent nature of cytotoxicity highlights the importance of considering both genomic potential and growth dynamics when assessing the biosafety of *B. cereus* group isolates for biotechnological applications. However, genome data alone cannot establish which compounds are actually produced under the tested conditions or confirm their role in the effect; further targeted studies will be necessary to clarify the mechanism and to complete the biosafety evaluation of strain al37.1^T^.

The comparative genomic analysis showed that the distribution of the metabolic components and virulence determinants was generally clade-independent, implying possible common activities of the isolates from the *B. al37.1^T^* group. Nevertheless, the overall genomic differences were noticeable. In a dark forest of *B. cereus s.l.* phylogeny, a relatively recent taxonomic nomenclature proposed by Carroll et al. in 2020 aimed to define genomospecies based on the pair-wise genome distances to disentangle intricate taxonomic relationships [27]. We leveraged the Btyper3 v3.4.0 [17] tool, designed specifically for genomospecies-wise taxonomic assignment of *B. cereus s. l.* genomes. While the final Btyper3-generated attributions, in general, finely fit the phylogenetic clades, they were not free from mislabelling. For instance, *B. altaicus* sp. nov. al37.1^T^ clade coupled with the larger subtree differing from *B. mycoides* since the DDH with DSM 2048^T^ regarding both ANI and DDH thresholds (Figure 3E). In addition, the novel nomenclature treated *B. nitratireducens* and *B. proteolyticus* [19] as members of the *B. mycoides* genomospecies IV [27]; however, the taxonomic estimates vividly support that they should be treated as separate species (Table 1). Another limitation lies in the absence of *B. hominis* [76] attributions representing a distinct species as well (Table 1). Therefore, despite a great advance in elucidating the intricate taxonomy of the *B. mycoides* subgroup of *B. cereus s. l.*, we still lack unambiguous distinction criteria. The limitations in current taxonomic classification presumably stem from the reliance on ANI distance-based clusters [27]. Since phylogenetic inferences are impacted by recombination-mediated gene transfer [77], ANI estimates [78], and gene composition [79] in species with high genetic mobility, considering the single criterion seems insufficient. An alternative classification could employ the usage of type strains coupled with a comprehensive analysis incorporating both morphological/physiological studies and genomic comparisons with preliminary selection of synthetic blocks devoid of HGT and recombination-subjected regions.

Therefore, not only did we propose a novel species, *B. altaicus*, distinct from its closest relative, *B. mycoides*, but also showed certain conserved functional loci in the subgroup of *B. cereus s. l.* The presence of enterotoxin-producing operons, coupled with the experimental evidence of cytotoxic effects of our strain, strongly suggests the necessity to assess the safety of strains close to *B. mycoides*. The results of this study provide perspectives for further research in the field. One direction lies in improving our understanding of the *B. cereus s. l.* taxonomy, which requires: (i) isolation of *Bacillus*-like species from various environments, (ii) obtaining high-quality genomes, (iii) performing a combination of genome-based taxonomic analysis, including whole genome comparison and marker loci-based typing, and (iv) assessing biochemical and morphological characteristics of these strains. Another key area is the functional description of *B. mycoides* group strains, i.e., (i) genome mining of functional loci related to virulence and metabolism, (ii) experimental testing of beneficial activities, such as insecticidal, antibacterial, and antifungal, and (iii) validation of cytotoxic effects, especially in already existing biopreparations.

## 4. Materials and Methods

### 4.1. Sampling Protocol

The soil sample covered 5 subsites over an area of approximately 1 × 1 m at a depth of 3–5 cm from a meadow in the Altai Republic, Russia (coordinates: 50.02369 N, 85.97772 E) in July 2022. During transportation, the samples were kept at room temperature, and subsequently stored in our laboratory at a temperature range of 4–10 °C.

### 4.2. Isolation of Bacilli Strains from Soil Samples

To isolate the strain, we started by suspending 0.2 g of soil in 2 mL of sterile water and vigorously stirring the mixture for 10 min using an FV-2400 vortex mixer (Biosan, Latvia). The homogenized sample was divided into 1 mL portions in Eppendorf tubes and heated in a water bath at 80 °C for 30 min to eliminate non-spore-forming organisms and vegetative *Bacillus* cells. Tenfold serial dilutions (10^−1^, 10^−2^, and 10^−3^) were prepared, and 200 μL of each suspension was plated onto T3 agar (composition per liter: tryptone 3 g, tryptose 2 g, yeast extract 1.5 g, NaH_2_PO_4_·H_2_O 6.9 g, MnCl_2_·4H_2_O 0.008 g, agar 15 g; pH adjusted to 6.8) [80] which favors sporulation. The inoculated plates were incubated for 72 h at 28 °C. Colonies with characteristic morphology were isolated and subcultured several times on fresh T3 agar to establish pure cultures. Pure cultures were stored long-term in a 25% (*v*/*v*) aqueous glycerol solution at −70 °C. For a detailed study, we chose the isolate that survived the heat treatment but, unlike typical survivors, did not form spores under our cultivation conditions, making it of particular interest.

### 4.3. Assessment of Optimal Growth Conditions

The optimal growth parameters of the strain, including pH, NaCl concentration, and temperature, were determined following the general approach of Antoniou et al., 1990 [81] with modifications. All experiments were performed in 2YT medium (16 g/L tryptone, 10 g/L yeast extract, 5 g/L NaCl) [13]. The pH of freshly prepared medium was 6.5, and the NaCl concentration was 0.5% (*w*/*v*). In the case of pH testing, the range from 4.0 to 11.0 was examined. Optical density (OD_600_) was measured at 30 min intervals over 48 h using a CLARIOstar Plus plate reader (BMG LABTECH, Ortenberg, Germany), and growth curves were plotted. For NaCl tolerance, a range from 0 to 15% (*w*/*v*) was tested using the same measurement protocol.

For temperature profiling, the strain was grown on solid 2YT medium [13] for 24 h in an air thermostat TSO-1/80 SPU (SKTB SPU, Smolensk, Russia), after which bacterial biomass was resuspended in liquid 2YT medium [13] to an OD_600_ of 0.2 and further diluted to 0.002. Cultures were incubated in a shaker incubator ES-20/60 (BioSan, Riga, Latvia) for 24 h at various temperatures (+5 to +45 °C). Samples were taken at 4, 8, and 24 h, and OD_600_ was measured using an IMPLEN DiluPhotometer (Implen GmbH, Munich, Germany). The final OD values at 24 h were used to construct the temperature growth profile. All experiments were performed in quadruplicate.

### 4.4. Morphological Description of the Strain

To describe standard colony morphology features, namely shape, elevation, optical properties (matte or glistening), color, surface appearance, and margin, as well as to analyze vegetative cell and spore morphology, the strain was cultivated on solid 2YT medium (16 g/L tryptone, 10 g/L yeast extract, 5.0 g/L NaCl) [13] at 30 °C. Colonies and vegetative cells were examined after 24 h of incubation, while spores were analyzed after 6 days, when the culture had entered the sporulation phase. Next, the culture was collected using a microbiological loop with a capacity of 10 μL and suspended in sterile deionized water on a slide, dried at room temperature, stained for 10–15 min with Coomassie Brilliant Blue Staining Solution (Bio-Rad Laboratories Inc., Hercules, CA, USA) using a Coplin jar, and rinsed with distilled water. The cells were examined under a light microscope (Carl Zeiss Axio Imager 2, Jena, Germany) at 1000× magnification.

Ultrastructural observations were performed by transmission electron microscopy (TEM) on a Jeol JEM-1400 transmission electron microscope (JEOL Corp., Tokyo, Japan) coupled to a Veleta CCD camera (Olympus-SIS, Münster, Germany). The preparations were applied onto copper grids previously covered with formvar–carbon support films (Electron Microscopy Sciences, Hatfield, PA, USA). The most common standard negative staining was carried out [82] with modifications to staining using 1% aqueous uranyl acetate for sample visualization [83,84]. The staining time was selected empirically and amounted to 1.5 min. After staining, 1 wash was performed with filtered mQ water, which was additionally autoclaved. TEM analysis was conducted on cultures grown for 24 h in liquid CCY (0.5 mM MgCl_2_ × 6H_2_O; 0.01 mM MnCl_2_ × 4H_2_O; 0.05 mM FeCl_3_ × 6H_2_O; 0.05 mM ZnCl_2_; 0.2 mM CaCl_2_ × 6H_2_O; 13 mM KH_2_PO_4_; 26 mM K_2_HPO_4_; 20 mg/L glutamine; 1 g/L acid casein hydrolysate; 1 g/L enzymatic casein hydrolysate; 0.4 g/L enzymatic yeast extract; 0.6 g/L glycerol; agar 20 g/L) medium [85] to observe spores and on cultures grown for 24 h in liquid 2YT medium [13] to observe vegetative cells.

Bacterial motility was evaluated by light microscopy. A 24 h culture grown in liquid 2YT medium [13] at 30 °C and 190 rpm in a shaker incubator ES-20/60 (BioSan, Riga, Latvia) was used. Five microlitres of the bacterial suspension were placed onto a clean microscope slide, gently covered with a coverslip held at ~45°, and lightly pressed so that no air bubbles remained and the suspension did not spread beyond the edges. The coverslip was sealed with clear nail polish (1–2 mm overlap on all four sides). All steps were performed rapidly to prevent drying. Motility was examined under oil immersion (×1000) with an Axio Imager 2 microscope (Carl Zeiss, Jena, Germany). The strain was classified as motile if directed movement was observed, even when exhibited by only a small proportion of cells [86]. If the cells were immobile or moved uniformly in a single direction, indicative of a convection current, the strain was classified as non-motile [87]. The experiment was conducted in three independent biological and two technical replicates.

### 4.5. Sequencing of 16S rRNA Locus

The procedure of DNA extraction, PCR amplification, and PCR product purification was previously described [63] with slight modifications. The primers for 16S amplification were 27F (5′-AGAGTTTGATCCTGGCTCAG-3′) and 1492R (5′-ACGGYTACCTTGTTACGACTT-3′) [88] with the amplification program consisting of an initial denaturation of 3 min at 94 °C, followed by 30 cycles of denaturation for 30 s at 95 °C, annealing for 30 s at 49 °C, and elongation at 72 °C for 1 min 30 s.

The sequencing procedure was carried out using the equipment available at the Core Centrum ‘Genomic Technologies, Proteomics and Cell Biology’ in ARRIAM and Evrogen company (Moscow, Russia).

### 4.6. Physiological and Biochemical Characterization

The enzymatic activities of the isolates were assessed using the GENIII MicroPlate microassay system (Biolog Inc., Hayward, CA, USA), which analyses the ability of bacteria to metabolize 71 carbon sources and resistance to 23 chemicals [89,90,91]. Fresh overnight cultures of isolates were tested as recommended by the manufacturer. Bacterial suspensions were prepared by removing bacterial colonies from the plate surface with a sterile cotton swab and emulsifying them into 10 mL of inoculating fluid (IF-A) to achieve 96–98% transmittance (T90) using a Biolog turbidimeter. The inoculated IF-A was dispensed into 96 wells of the Biolog GEN III microplate (100 μL per well) using a multichannel repeating pipettor. The built-in positive and negative controls supplied with the plate were used as recommended by the manufacturer. The microplate was incubated at 27 °C in a thermostat (BINDER, Neckarsulm, Germany) for 24 h, and the results were read on a semi-automated MicroStation reader (Biolog Inc., USA).

The urease and catalase activities of the strains were studied using ready-to-use kits (NICF company, Saint Petersburg, Russia) according to the manufacturer’s instructions. *Escherichia coli* 756 was included as a positive control for the catalase test and as a negative control for the urease test, while *Phyllobacterium zundukense* Tri-48 served as a positive control for urease. Each assay was performed in duplicate.

### 4.7. Chemotaxonomic Analysis

The fatty acid (FA) profile of cell lipids was analyzed essentially as described in [92,93]. Briefly, the cells were pelleted by centrifugation, transferred to a glass–glass homogenizer, and disrupted in a chloroform–methanol (10 mL, 2:1, *v*/*v*) mixture [94]. The chloroform fraction containing total cell lipids was extracted and evaporated to dryness by a stream of argon. For transmethylation, lipid extracts were incubated in anhydrous methanol supplemented with 2% (*v*/*v*) H_2_SO_4_ at 80 °C for 1.5 h under argon atmosphere with continuous agitation. Heptadecanoic acid (C17:0) (Fluka, Buchs, Switzerland) served as an internal standard. The resulting fatty acid methyl esters were examined by gas chromatography [93]. Separation and identification were based on retention times of reference standards (Sigma, St. Louis, MO, USA) and through analysis of characteristic mass spectra acquired using an Agilent 7890 gas chromatograph fitted with a 30 m HP5MS UI capillary column coupled with an Agilent 5970 mass-selective detector (Agilent, Santa Clara, CA, USA).

### 4.8. Genomic DNA Extraction, Quality Control, and Library Preparation

Total genomic DNA for Oxford Nanopore Technologies (Oxford Nanopore Technologies Ltd., Oxford, UK) sequencing was extracted using the MagBeads FastDNA Kit for Microbiome (MP Biomedicals, Santa Ana, CA, USA) according to the manufacturer’s manual.

Library preparation for sequencing was performed according to the manufacturer’s manual using the Native Barcoding Kit 24 V14 (Oxford Nanopore Technologies Ltd., Oxford, UK), the NEBNext Companion Module for Oxford Nanopore Technologies Ligation Sequencing (New England Biolabs, Ipswich, MA, USA), NEBNext Quick Ligation Reaction Buffer, and Blunt/TA Ligase Master Mix (New England Biolabs, Ipswich, MA, USA). Sequencing was carried out on a MinION Mk1B device (Oxford Nanopore Technologies Ltd., Oxford, UK) with an R10 flow cell (Oxford Nanopore Technologies Ltd., Oxford, UK).

DNA for sequencing on the Illumina NovaSeq X (Illumina Inc., San Diego, CA, USA) platform was extracted and qualitatively assessed according to the protocol described in our previous research [95]. Briefly, we first incubated the strain in liquid Spizizen medium ((NH_4_)_2_SO_4_ 2g/L; KH_2_PO_4_ 6 g/L; Na_3_C_6_H_5_O_7_ 1 g/L; MgSO_4_ × 7H_2_O 0.2 g/L; glucose 0.5%; K_2_HPO_4_ × 3H_2_O 18.3 g/L; tryptone 20g/L; yeast extract 5 g/L) [12,13] overnight and then washed the cells from the medium with buffer (EDTA 0.01M, NaCl 0.15 M, pH 8.0). In the next step, we incubated the samples for 60 min at +37 °C with the addition of the above-mentioned buffer, ribonuclease A (10 mg/mL), lysozyme (20 mg/mL), and mutanolysin (1 mg/mL). Next, we incubated the sample with proteinase K (600 U/mL) for 10 min at +37 °C and 10% sodium dodecyl sulfate for 10 min at +65 °C. A modification of the described protocol lay in adding Protein Precipitation Solution (Qiagen, Venlo, The Netherlands) to the samples instead of using a phenol and chloroform mixture. Subsequent steps of DNA precipitation and quality control remained unchanged.

### 4.9. Genome Assembly and Annotation

Whole genome sequencing was conducted on the Illumina NovoSeq X platform in paired-end mode with 2 × 100 bp reads, performed by Novogene Co., Ltd. in Beijing, China. Raw sequencing reads underwent quality control using FastQC v0.12.1 (https://www.bioinformatics.babraham.ac.uk/projects/fastqc/ (accessed on 17 April 2024)) [96] and fastp v0.23.2 [97]. Long reads were obtained on the Oxford Nanopore platform, followed by basecalling using Dorado software v0.8.3 (https://github.com/nanoporetech/dorado (accessed on 17 November 2024)) and quality assessment using the RabbitQCPlus v2.2.9 [98] utility. The draft genome was then assembled using Flye v2.9.5-b1801 [99] with three iterations, followed by consensus polishing with Medaka v2.0.1 (Oxford Nanopore Technologies Ltd., UK) and short reads-based polishing with BWA 0.7.18-r1243-dirty [100], SAMtools v1.21-21-gaa8cb59 [101], and Pilon v1.24 [102] with three polishing rounds specified. The assembly’s quality was subsequently evaluated with QUAST v5.2.0 [103], BUSCO v5.4.2 [15], and CheckM v1.2.2 [104]. Next, we picked the 10 most similar genome assemblies belonging to the order of Bacillales from the NCBI RefSeq [16] database by applying Average Nucleotide Identity (ANI), which was calculated using fastANI v1.33 [105]. These genomes were used to construct a gene-prediction model for Prokka v1.14.6 [106] with an additional in-house database incorporating non-redundant protein sequences from BUSCO [15] reference proteomes of the order of Bacillales for annotation 4.10. Genome-wise Taxonomy Classification.

The data processing was performed with Python v3.12. The plots were generated with the ggplot2 v3.3.5 [107] package for R v4.5.0.

### 4.10. Taxonomic Classification

To find the closest reference strains, we leveraged the available high-quality genomes from the NCBI RefSeq [16] database and selected 300 closest representatives according to the ANI metrics reported by fastANI v1.33 [105]. We also included the reference strain 168 of *B. subtilis* as the outgroup. The metadata of the respective strains was then taken from the NCBI BioSample [108] database. The core phylogeny representing general relationships between the genomes studied was obtained using a pangenome-based approach with core SNPs (Single Nucleotide Polymorphisms) extraction. We first prepared the pangenome in re-annotated assemblies with Panaroo v1.5.0 [109] and retrieved core SNPs using SNP-sites v2.5.1 [110]. Next, we detected the best evolutionary model by inspecting the BIC (Bayesian Information Criterion) estimates provided by the ModelTest-NG v0.1.7 [111] software. The reference ML (Maximum Likelihood) tree was constructed utilizing RAxML-NG v1.2.2 [112]. The tree was visualized with the ggtree v1.16.6 [113] package.

For retrieving the 16 rRNA loci, we applied Barrnap v0.9 (https://github.com/tseemann/barrnap (accessed on 17 April 2024)) and extracted the longest sequences from each genome in the case of fragmented assemblies. The above-mentioned ML-grounded approach was applied to reconstruct phylogeny. The topological comparisons with the reference core SNPs tree were performed with the quartered distance metrics with the tqDist v1.0.2 library [114]. The recombination-to-mutation ratio (r/m rate) was evaluated using ClonalFrameML v1.12 [115] on the concatenated core genes alignment. Next, we calculated DDH (digital DNA-DNA Hybridization) scores utilizing the GGDC [116] service relative to our isolate and the reference *B. mycoides* strain DSM 2048. To reveal the taxonomic attribution of the included strains within the *B. cereus sensu lato* group, we used the Btyper3 v3.4.0 [17].

To perform MLST analysis, we used the seven-gene MLST scheme implemented in the PubMLST [117] database. We collected all allelic variants for each gene and the total clonal complex classification obtained using Btyper3 v3.4.0 [17] and mlst v2.23.0 (https://github.com/tseemann/mlst (accessed on 17 April 2024)). To obtain loci-based phylogenetic inferences, we downloaded the MLST gene sequences from the PubMLST resource, followed by genome mining of these loci in pangenomic sequences utilizing MMseqs2 v.14.7 [118]. For each set of sequences, individual evolutionary models were identified with ModelTest-NG v0.1.7 [111], followed by ML reconstruction as described above with model partitions for a concatenated alignment. A single locus-based phylogeny was reconstructed on *gyrB* according to the above-described pipeline. The inferences were then compared topologically with the reference tree with the tqDist v1.0.2 library [114].

### 4.11. Analysis of Functional Loci

Each assembly from the genomic dataset was scanned for genes encoding insecticidal toxins with BtToxin_Digger v1.0.10 [119], IDOPS v0.2.2 [120], and CryProcessor v1.0 [121]. The virulence potential was assessed by identifying the enterotoxins’ repertoire with BTyper3 v3.4.0 [17] and identifying homologs of the known virulence determinants with MMseqs2 v.14.7 [118] against the VFDB database (http://www.mgc.ac.cn/VFs/ (accessed on 21 February 2025)) [24] with 70% thresholds of identity and mutual coverage. Sortases were predicted using the SortPred v1.0 [122] software. The loci responsible for bactericidal, fungicidal, and other activities were predicted with DeepBGC v0.1.30 [23] and antiSMASH v7.1.1 [123]. Since the DeepBGC-based predictions are often short in length, we selected the thresholds of at least 3 CDS in the region and not less than 3000 b.p. in length as the minimal values related to the antiSMASH results.

### 4.12. Cytotoxicity Assay

The cytotoxicity of *B. altaicus* al37.1^T^ culture against the human pancreatic cancer cell line PANC-1 (ATCC CRL-1469, obtained from the shared research facility “Vertebrate cell culture collection” of the Institute of Cytology of the Russian Academy of Sciences (St. Petersburg, Russia) was assessed as follows. The strain was cultivated in liquid 2YT medium [13] at 28 °C and 190 rpm. After 7, 16, and 29 h of cultivation, the culture suspensions were filtered through 0.45 µm PVDF syringe filters. The PANC-1 cells were plated in 96-well culture plates in 90 µL of DMEM medium (25 mM D-glucose, 4.0 mM L-glutamine, 1.0 mM sodium pyruvate, phenol red; Servicebio, Wuhan, China) supplemented with 10% FBS (HyClone, Logan, UT, USA), 100 U/mL penicillin, and 100 µg/mL streptomycin (Capricorn Scientific, Ebsdorfergrund, Germany). The filtrates from each cultivation time point were added to the cells at final dilutions of 1:10, 1:100, and 1:1000. Control wells contained PANC-1 cells in complete culture medium supplemented with 2YT at the same final concentrations (1:10, 1:100, 1:1000) as used for the bacterial filtrates, so that the total volume matched that of the experimental wells. Cells were incubated at 37 °C and 5% CO_2_ for 24 h. The viability of the cells was determined following the protocol reported in Kumar et al., 2018 [124]. In short, 10 µL of MTT solution in PBS (5 mg/mL) was dispensed into each well, and the cells were incubated at 37 °C under 5% CO_2_ for 4 h. Subsequently, 100 µL of SDS–HCl solution (10% SDS, 0.01 N HCl) was added, and the plates were further incubated for 18 h. Optical density was recorded at 570 nm and normalized by subtracting the readings at 620 nm.

All measurements were performed in four technical and three biological replicates. To compare multiple groups, the one-way analysis of variance (ANOVA) method (from rstatix package v0.7.2, https://github.com/kassambara/rstatix (accessed on 17 May 2025)) was applied to the optical density measurements, followed by the emmeans post hoc test using the emmeans R package v1.10.7 (https://github.com/rvlenth/emmeans (accessed on 17 May 2025)). Differences were considered statistically significant at a *p*-value less than 0.05. To plot the data, the measurements were divided by the median value of the control group. The normalized median values with interquartile ranges have been plotted. All calculations were performed using R language v4.4.3 [125].

## 5. Conclusions

To sum up, although we are still far from a coherent taxonomic system describing bacterial species as separate units, the advances in multidisciplinary approaches would certainly contribute to the development of an ideal classification. It is especially important to cover distinct geographic regions to broaden the scope of microbial diversity in nature. To this end, we presented a comprehensive description of the al37.1^T^ strain isolated from a poorly studied Altai region. A combination of morphological description and genome-based inferences allowed us to allocate a novel species—*Bacillus altaicus*, moderately but significantly differing from the closest counterpart in the *B. cereus s. l.* group, i.e., *B. mycoides*. We demonstrate that the reliance on a single criterion is certainly insufficient to describe novel species. Having performed a comprehensive genomic analysis of the whole group, we revealed both inconsistencies yet to be resolved and a uniform presence of biosynthetic gene clusters constituting a metabolic core of the whole group. Given the overwhelming presence of complete operons controlling the production of enteric toxins based on the genomic data and the observed cytotoxic effect on the *B. altaicus* al37.1^T^, we could raise a healthcare issue underpinning the necessity to conduct an accurate assessment of the toxigenic potential when introducing novel biocontrol agents. All things considered, the current study emphasizes the potency of applying multiple approaches in describing new bacterial species and the need for further comparative genomic analysis for identifying core functional markers delineating the biological properties of distinct species and the complex groups.

## Figures and Tables

**Figure 1 ijms-26-09517-f001:**
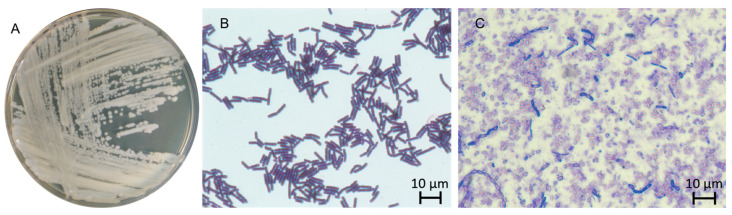
Morphology of *B.* sp al37.1. (**A**) Colonies after 24 h cultivation on 2TY medium; (**B**) Gram-stained *B.* sp. al37.1^T^ spore culture on day 6 stained with Coomassie Brilliant Blue; (**C**) 1spore culture on day 6 stained with Coomassie Brilliant Blue; magnification ×1000.

**Figure 2 ijms-26-09517-f002:**
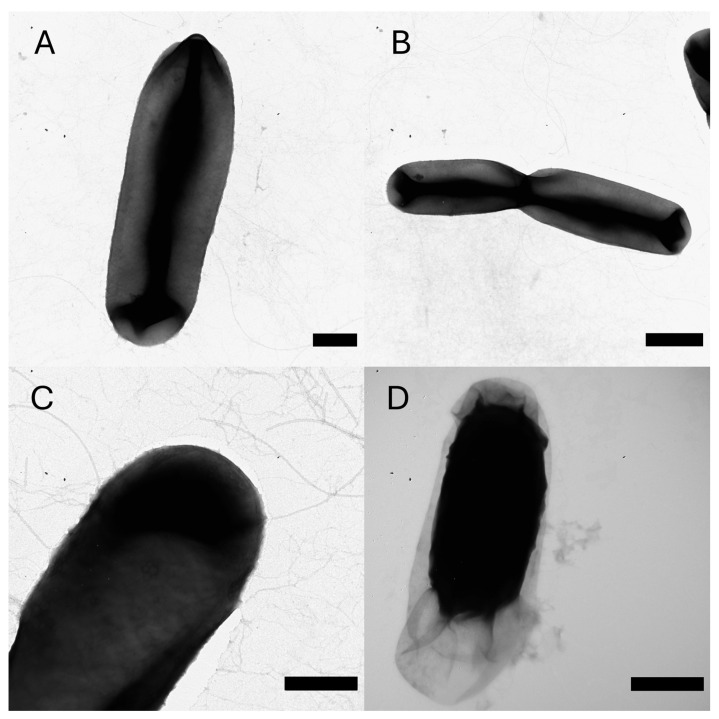
Transmission electron micrographs of *B*. sp. al37.1 showing cell morphology, filamentous structures, and sporulation. (**A**–**C**) represent vegetative cells with peritrichous filaments. (**D**) Spore with a surrounding endosporium. Scale bars: (**A**,**C**,**D**)—500 nm; (**B**)—1 µm.

**Figure 3 ijms-26-09517-f003:**
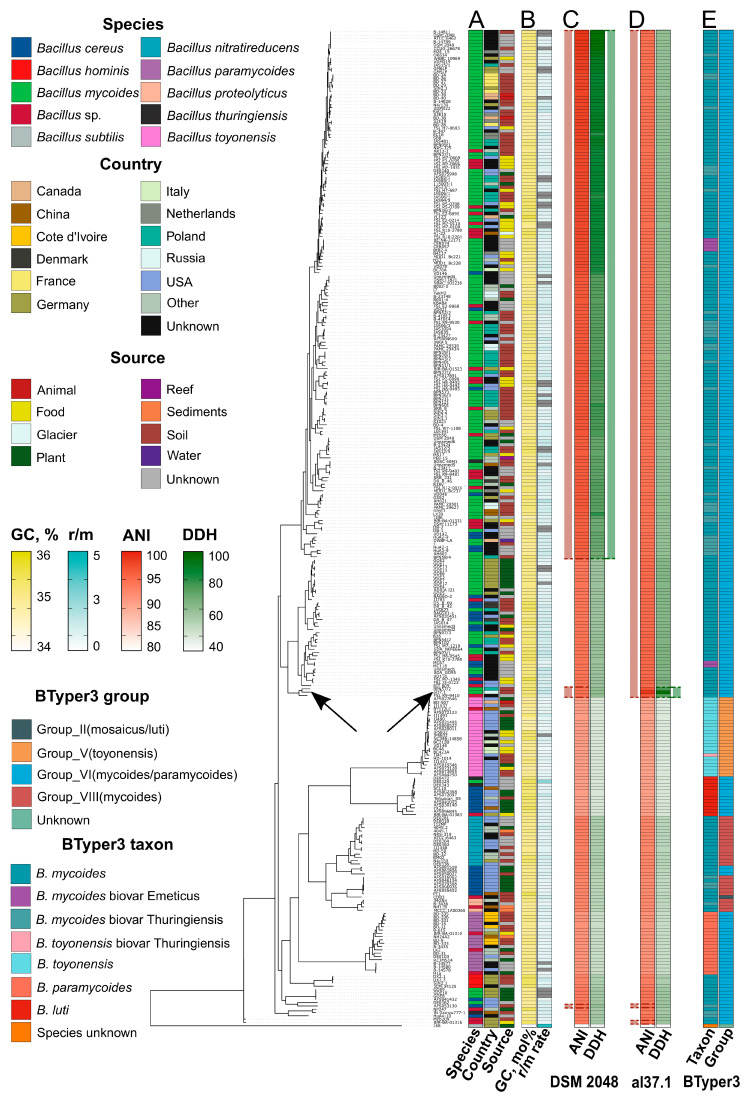
Taxonomic description of strain al37.1^T^ according to genomic characteristics in comparison with reference genomes. The presented phylogeny was reconstructed using the ML (Maximum Likelihood approach) on core SNPs (Single Nucleotide Polymorphisms). (**A**) The general metadata of the 300 closest reference genomes attributed to the Bacillales order. Presented are the claimed species attribution in the NCBI resource, the country of origin, and the isolation source. (**B**) Base genomic properties of the analyzed genomes. The color is proportional to the values, namely, GC-content and r/m (recombination-to-mutation) rates. In case the evaluation of r/m on branches was unavailable, the respective tiles are colored gray. (**C**) Comparisons of genomic similarity relative to the reference *B. mycoides* strain (DSM 2048^T^) and isolate al37.1^T^ (**D**). Two similarity metrics, namely, ANI (Average Nucleotide Identity) and DDH (digital DNA-DNA Hybridization), are illustrated. The color is proportional to the similarity scores. Red and green stripes adjacent to the strips demarcate the species threshold according to the established thresholds (95% ANI and 70% DDH), constituting the same species unit with DSM 2048 and al37.1^T^ strains, respectively. (**E**) Taxonomic classification of the isolates according to Btyper3 v3.4.0 [17] results.

**Figure 5 ijms-26-09517-f005:**
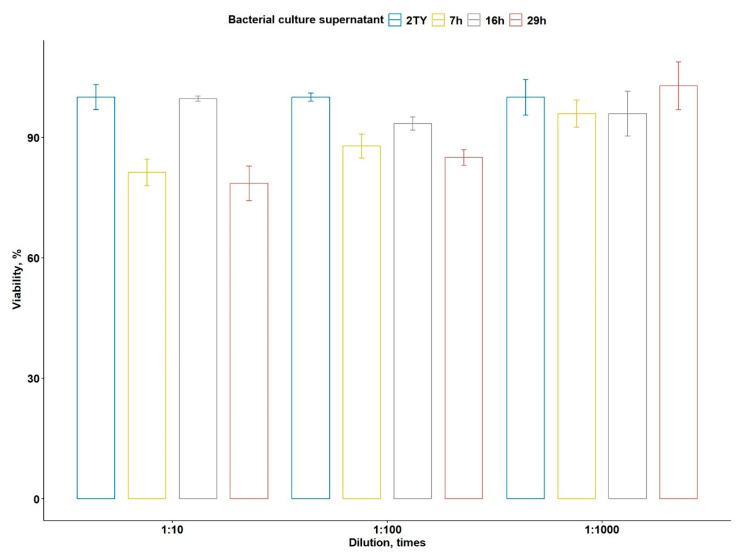
Cytotoxic effects of *Bacillus altaicus* al37.1^T^ culture supernatants on PANC-1 cells after 24 h incubation. Supernatants were collected at different cultivation stages (7, 16, and 29 h) and applied at dilutions of 1:10, 1:100, and 1:1000. The viability of the control samples was taken as 100%. Bars denote the median values divided by median values of the control group, with error bars denoting the interquartile range.

**Table 1 ijms-26-09517-t001:** Key taxonomic characteristics of the type strains included in the dataset. Sequence comparisons (DDH, ANI, 16S rRNA) are performed relative to the *B. mycoides* type strain DSM 2048^T^ and *B. altaicus* sp. nov. strain al37.1^T^.

Feature/Strain	DSM 2048^T^	al37.1^T^	NCIMB 14858 (BCT-7112^T^)	4049^T^	MCCC 1A00365 (TD42^T^)	NH24A2^T^	JCM 34125 (BML-BC059^T^)	168
Assembly	GCF_029024805.1	GCF_042136375.1	GCF_000496285.1	GCF_001884135.1	GCF_001884065.1	GCF_001884235.1	GCF_018332515.1	GCF_964017255.1
Species	*B. mycoides*	*Bacillus altaicus* sp. nov	*B. toyonensis*	*B. nitratireducens*	*B. proteolyticus*	*B. paramycoides*	*B. hominis*	*B. subtilis*
BTyper3 taxon	*B. mycoides*	*B. mycoides*	*B. toyonensis*	*B. mycoides*	*B. mycoides*	*B. paramycoides*	*B. mycoides*	Unknown
Genome length, bp	5,639,253	5,397,466	5,025,419	5,480,833	5,847,531	5,444,255	5,192,534	4,215,606
Number of genes	5817	5458	5198	5554	5950	5501	5239	4332
GC content, %	35.4	35.5	35.6	35.3	35.2	35.2	35.5	43.5
ANI (DSM 2048^T^)	-	95.3	90.8	93.9	93	91.5	93.7	78.1
DDH (DSM 2048^T^)	-	61.6	41.5	53.4	50.3	45.7	53.2	34.2
16S rRNA similarity (DSM 2048^T^)	-	99.9	99.4	99.2	99.6	99.4	99.7	93.7
ANI (al37.1^T^)	95.3	-	90.2	93	92.5	91.7	94	76
DDH (al37.1^T^)	61.6	-	40.1	49.7	48.6	46.1	54.1	31.7
16S rRNA similarity (al37.1^T^)	99.9	-	99.5	99.4	99.7	99.6	99.8	93.7
*gyrB* similarity (al37.1^T^)	97.77	-	90.15	91.19	91.14	93.99	95.9	70.98
*gyrB* similarity (DSM 2048^T^)	-	97.77	91.08	91.65	91.76	93.57	95.33	70.67
MLST complex	116	Unknown	111	769	765	780	3200	1

**Table 2 ijms-26-09517-t002:** Carbon sources utilized by *B. altaicus* al37.1^T^ as determined by the Biolog GEN III microplate assay.

Carbohydrates and Derivatives	Alcohols	Amino Acids	Organic Acids and Derivatives	Other Compounds
D-Maltose D-Trehalose D-Cellobiose D-Turanose N-Acetyl-β-D-Mannosamine β-Methyl-D-Glucoside N-Acetyl-D-Glucosamine N-Acetyl-D-Galactosamine α-D-Glucose D-Mannose D-Fructose D-Galactose L-Rhamnose L-Fucose D-Fucose D-Glucose-6-PO_4_ D-Fructose-6-PO_4_ 3-Methyl Glucose D-Salicin	D-Arabitol myo-Inositol Glycerol	D-Serine L-Arginine L-Histidine L-Alanine L-Aspartic Acid L-Glutamic Acid L-Serine	L-Galactonic Acid Lactone D-Gluconic Acid D-Glucuronic Acid Glucuronamide Quinic Acid D-Saccharic Acid p-Hydroxy-Phenylacetic Acid Methyl Pyruvate D-Lactic Acid Methyl Ester L-Lactic Acid α-Ketoglutaric Acid L-Malic Acid Bromo-Succinic Acid α-Hydroxy-Butyric Acid β-Hydroxy-D, L-Butyric Acid α-Keto-Butyric Acid Propionic Acid Acetic Acid Formic Acid	Inosine Glycyl-L-Proline

**Table 3 ijms-26-09517-t003:** Comparative phenotypic characteristics of the novel *B. altaicus* al37.1^T^ and the reference strains *B. mycoides* DSM 2048^T^ and *B. toyonensis* BCT-7112^T^. Data for *B. mycoides* DSM 2048^T^ (rhyzoidal colony, parasporal crystal, cellular fatty acids, carbon source utilization) were taken from Guinebretière et al., 2013 [18], and temperature/pH/NaCl tolerance ranges, catalase/urease test results, and cell size from Liu et al., 2017 [19]. Data for *B. toyonensis* BCT-7112^T^ were taken from Jiménez et al., 2013 [20]. +, positive; −, negative; ND, not determined.

Characteristic	*B. altaicus* sp. nov. al37.1^T^	*B. mycoides* DSM 2048^T^	*B. toyonensis* BCT-7112^T^
Cell length (μm)	1.5–4.3	3.0–5.0	3.0–4.4
Cell width (μm)	0.6–1.8	>1	>1
Rhyzoidal colony	−	+	−
Parasporal crystal	−	−	−
Temp range (°C)	+20–40	+15–40	+10–45
Optimal temp (°C)	+30	+30	+35
pH range	6–9	5–9.5	5–9.5
Optimal pH	6.5	8	6.5
NaCl range (%, wt/vol)	0–1%	0–4	0–5
Optimal NaCl concentration (%, wt/vol)	0.5	1	0
Catalase test	+	+	+
Urease test	−	−	−
**Cellular fatty acids (% of total)**			
12:0	4.3	2.7	N/D
14:0	9.6	3.7	3.2
16:0	34.8	15.6	5.6
18:0	7.9	1.6	N/D
18:1	15.2	N/D	N/D
**Carbon source utilization**			
Glycerol	+	−	−
D-Mannose	+	−	−
D-Salicin	+	+	+
D-Cellobiose	+	Weakly positive	−
Sucrose	−	+	+
D-Trehalose	+	+	+
D-Turanose	+	−	+
D-Fructose	+	+	+
D-Maltose	+	+	+
Gentiobiose	−	−	−
L-Rhamnose	+	+	−
D-Sorbitol	−	+	−
D-Melibiose	−	+	−
D-Raffinose	−	+	−
D-Fucose	+	+	−
L-Fucose	+	+	−
D-Arabitol	+	+	−
D-Galactose	+	−	−

## Data Availability

The raw genome sequencing using Illumina HiSeq X and Oxford Nanopore were deposited in the NCBI SRA database (SRR35002070 and SRR35002069). The metadata is available in the BioSample (SAMN43771369) and BioProject (PRJNA1161141) entities. The annotated genome was deposited in the NCBI GeneBank database under GCF_042136375.1.

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
