# Peer review of "Proposal of *Bacillus altaicus* sp. nov. Isolated from Soil in the Altai Region, Russia"

_ijms, 2025, doi:10.3390/ijms26199517_

Round 1
Reviewer 1 Report
Comments and Suggestions for Authors
The manuscript is generally well written, with a clear research logic. This study reported the isolation and identification of a novel strain, Bacillus altaicus sp. nov., from soil samples collected in the Altai region of Russia. Through genome sequencing, phylogenetic analysis, and comparisons of phenotypic, physiological, and chemotaxonomic characteristics, the authors demonstrates that this strain is clearly distinct from B. mycoides and related species, supporting its proposal as a new species. In addition, the study further reveals unique features in its metabolic capacity, fatty acid composition, and secondary metabolite biosynthetic gene clusters, and also identifies potential insecticidal activity as well as the presence of toxin-associated genes conferring cytotoxicity toward human cells. Overall, this work contributes new insights into the diversity of the Bacillus genus and suggests potential applications in agriculture and biotechnology, although its safety profile or weaknesses requires further evaluation. Moreover, after carefully reviewing the manuscript, I still have some concerns and questions that the authors are encouraged to address. The concerns are as follows:
- In proposing the new species, the manuscript primarily relies on 16S rRNA sequence similarity to preliminarily associate strain al37.1T with Bacillus mycoides. However, since 16S rRNA has limited resolution within the B. cereus group, reliance on this single marker gene may be insufficient to support species-level identification. Although the authors have provided genomic evidence such as ANI and dDDH, it is recommended to include multilocus sequence analysis (e.g., gyrB, rpoB, recA, atpD and other housekeeping genes) to further strengthen the robustness of phylogenetic inference and the validity of the new species proposal.
- The manuscript reports a mild cytotoxic effect of the strain on PANC-1 cells (with cell viability reduced by approximately 19–22%) and identifies the presence of the nheABC and hblABC operons. It is suggested that the authors emphasize more clearly in the discussion that the causal relationship between this observation and specific metabolites has not been established, and that future mechanistic studies will be needed to confirm this link and better evaluate the biosafety of the strain.
- The ecological significance of microbial diversity in the Altai region is not sufficiently discussed. A fuller and more reasoned discussion of the environmental gradients and their potential role in shaping B. altaicus would enhance the novelty of the study.
- In the sentence “Nevertheless, the presence of the operons per se does not imply the synthesis of these toxins [65]; thus, both laboratory experiments and genomic surveys is needed when implementing the B. mycoides group representatives in bioformulations to ensure safe usage.” (lines 453–456), there is a grammatical error. The phrase “…both laboratory experiments and genomic surveys is needed…” should be corrected to are needed.
- In the phrase “in agreement with the genomic predictions., Taken together,” (line 475), there is redundant punctuation; the comma should be removed or replaced with a period.
- In the reference section, multiple instances such as “Bacillus {thuringiensis}” appear due to formatting issues. These should be corrected to the standard italic form Bacillus thuringiensis.
Author Response
We are grateful to the reviewer for the overall positive assessment of our work and for the useful suggestions and comments that assisted in the refinement of the manuscript. We carefully addressed all the issues mentioned and conducted the loci-based phylogenetic analysis that confirmed our main findings. We hope that the amended research meets all the requirements. Below, please find our point-to-point answers.
- In proposing the new species, the manuscript primarily relies on 16S rRNA sequence similarity to preliminarily associate strain al37.1T with Bacillus mycoides. However, since 16S rRNA has limited resolution within the B. cereus group, reliance on this single marker gene may be insufficient to support species-level identification. Although the authors have provided genomic evidence such as ANI and dDDH, it is recommended to include multilocus sequence analysis (e.g., gyrB, rpoB, recA, atpD , and other housekeeping genes) to further strengthen the robustness of phylogenetic inference and the validity of the new species proposal.
Response: We thank the reviewer for raising an important question. While the pangenome-wise phylogeny is, at a certain point, an extended version of the MLST-based phylogeny, since it encompasses all the housekeeping genes, performing the established analysis with the accepted scheme is indeed a common practice in taxonomic studies. We used the seven-gene MLST scheme for Bacillus cereus and collected the allelic profiles. We reconstructed the respective phylogeny as well. The allele scheme of our strain substantially differed from the conventional B. mycoides clade. In addition, we used gyrB sequences and obtained the phylogenetic tree accordingly. Both MLST- and gyrB-based analyses showed that the B. altaicus clade was retained. These findings emphasize that the proposed species unit is correct. We provided the results (l. 183-196, Figure S1, Tables S3-S4) and extended the Discussion section (l. 392-398) accordingly.
- The manuscript reports a mild cytotoxic effect of the strain on PANC-1 cells (with cell viability reduced by approximately 19–22%) and identifies the presence of the nheABC and hblABC operons. It is suggested that the authors emphasize more clearly in the discussion that the causal relationship between this observation and specific metabolites has not been established, and that future mechanistic studies will be needed to confirm this link and better evaluate the biosafety of the strain.
Response: We thank the reviewer for this valuable remark. We have revised the discussion to clarify that the link between the observed cytotoxicity and specific metabolites or toxins has not been experimentally confirmed and that further mechanistic studies are required to verify this relationship and fully assess the biosafety of the strain (l. 499-502).
- The ecological significance of microbial diversity in the Altai region is not sufficiently discussed. A fuller and more reasoned discussion of the environmental gradients and their potential role in shaping B. altaicus would enhance the novelty of the study.
Response: We fully agree with the reviewer that analyzing the ecology of the strain could explain its adaptation and, eventually, the emergence of the species. Unfortunately, a few works describe the region. Bacterial diversity in Altai is mainly studied in specific biomes, while the soil-borne bacteria remain largely unknown. However, we found that environmental gradients are present in the articles describing phenotypic differences of higher plants from distinct geographic zones. According to these data, there are visible gradients of nutrients that could contribute to the emergence of novel bacterial species. Nevertheless, the exact metabolites were not characterized; thus, we are unable to find evident relationships between the morphological and biochemical properties of our strain. Therefore, further research is needed to uncover the ecological factors shaping the microbial diversity and species content, which we mentioned in the Discussion section (l. 411-431).
- In the sentence “Nevertheless, the presence of the operons per se does not imply the synthesis of these toxins [65]; thus, both laboratory experiments and genomic surveys is needed when implementing the B. mycoides group representatives in bioformulations to ensure safe usage.” (lines 453–456), there is a grammatical error. The phrase “…both laboratory experiments and genomic surveys is needed…” should be corrected to are needed.
Response: The sentences were fixed, as suggested (l. 476).
- In the phrase “in agreement with the genomic predictions., Taken together,” (line 475), there is redundant punctuation; the comma should be removed or replaced with a period.
Response: We apologize for the misprint. Erroneous comma was removed (l. 496).
- In the reference section, multiple instances such as “Bacillus {thuringiensis}” appear due to formatting issues. These should be corrected to the standard italic form Bacillus thuringiensis.
Response: We thank the reviewer for mentioning this issue and apologize for the incorrect formatting stemming from the reference manager. We carefully checked the reference list and removed invalid characters.

Reviewer 2 Report
Comments and Suggestions for Authors
Review of the article 'Proposal of Bacillus altaicus sp. nov. isolated from soil in the Altai region, Russia' by Anton E. Shikov, Maria N. Romanenko, Fedor M. Shmatov, Mikhail V. Belousov, Alexey E. Solovchenko, Olga Chivkunova, Grigoriy K. Savelev, Irina G. Kuznetsova, Denis S. Karlov, Anton A. Nizhnikov, Kirill S. Antonets
The article is devoted to the isolation and description of the new bacterial species Bacillus altaicus. However, the work presented raises serious concerns regarding its content, as well as the quality and layout of the illustrations.
Introduction:
The introduction clearly demonstrates knowledge of the context of Bacillus cereus sensu lato and reveals the relevance of research into microbial diversity in Altai. However, the presentation's structure is overly detailed regarding the classification and evolution of the group, and lacks a clear formulation of the scientific problem and the novelty of this particular study. Information from different areas is intertwined in the text, and the transitions between paragraphs are sometimes inconsistent, which reduces the narrative's logic and complicates its perception. Advantages include a broad review of the literature and justification of the choice of subject, but a clear outline of the work's tasks and hypotheses is necessary to improve its structure.
Materials and Methods
- Figure 8 does not convey any information and should be deleted.
- Isolation of bacilliform strains from soil samples is possible, but not recommended because 'bacilliform' means 'rod-shaped' and does not refer to belonging to the Bacillus genus. Scientific articles usually use 'bacilli' or 'Bacillus-type bacteria'.
- 'Approximately 10 μl of the 547 culture was collected with the tip of a microbiological needle' is an incorrect statement, as the biomass of colonies collected from agarised media cannot be measured in volume.
- Selection criteria. The section does not clearly describe the criteria for selecting strains for further analysis, and it is unclear on what basis objects were selected for detailed study from all isolates.
- Representativeness and Replicates: The number of biological and technical repetitions performed for each type of experiment is not specified, which makes it difficult to assess the statistical reliability of the obtained results.
- Lack of control. Most of the methods described, for example those used to determine physiological properties and conduct cytotoxic testing, make no mention of the use of positive and negative control groups.
- Protocol detail. Some methods are described too briefly, providing insufficient information to reproduce individual procedures. There are also no references to generally accepted standards or detailed parameters for performing analyses, and often only the basic steps are given.
For instance, the descriptions of sample preparation for microscopy and the reagents used are superficial. Specific concentrations, incubation times and temperatures at individual stages are not provided, making it impossible to repeat the procedure in another laboratory. The parameters for fixing and dehydrating samples for electron microscopy are not specified, nor are the imaging modes (e.g. accelerating voltage and contrast details).
Similarly, the sequences of the control and reference strains used to compare morphological features and biochemical tests are not provided — they are simply mentioned without any identifiers or information about where they came from.
- Sample storage. The storage conditions of the samples and strains under study at each stage are not fully disclosed, which may affect their condition and the results of the experiments.
- Statistical processing. The description of the statistical data processing methods used is minimal. Details of the approaches employed to analyse the significance of differences and interpret the results should be provided.
- Equipment and reagents. Information about the equipment, commercial kits and reagents used is lacking; it is advisable to add details such as manufacturers, batches and specific device models.
Results:
1) Figure 1 is not informative and should be moved to the supplementary materials.
2) Microscopy quality. The quality of the microscopic images presented is extremely poor and does not meet the current publication requirements of leading international journals. In particular: The images shown in Figure 2 are blurry and have poor contrast. Additionally, the TEM images lack detail in the membranes, spores and surface structures, which are essential for differentiation within the Bacillus cereus sensu lato group. The scale indicators are illegible, which makes it impossible to assess the size of the cell structures objectively.
Illustrations 5 and 6 are unclear and uninformative. Figures 5 and 6, which illustrate metabolic potential (biosynthetic gene clusters, BGCs) and pathogenicity, are extremely poorly designed, making it difficult to interpret the results. The legends are confusing and the visual elements (colours, gradients and axis labels) are not clearly explained or correspond to the text description. The data structure remains unclear.
Figure 5 shows the distribution of BGCs, but even upon detailed examination, it is impossible to understand it: the graph is overloaded with numerous small elements, preventing a clear picture of the differences between strains from emerging.
Figure 6 shows the presence of virulence factors in the form of a heatmap, which makes it practically impossible to distinguish between clusters and compare strains.
Both figures are unsuitable for printing or electronic viewing: the elements are small, the captions overlap and the visualised values (e.g. number of genes, colour intensity) lack intuitive explanations.
Consequently, rather than helping, the illustrative material greatly hinders the perception and critical evaluation of the comparative data provided by the author.
Discussion
Both figures are not suitable for printing or electronic viewing due to their scale and format: the elements are small and the captions overlap.
While the 'Discussion' section of the article is well structured and meets the basic requirements of microbiology journals, there are some aspects that could be improved. It opens with a brief summary of the main results, places them in the context of previous studies and focuses on novelty — an approach that is consistent with the recommendations of major scientific publishers. The interpretations of the results are detailed: the genomic, physiological and biochemical differences of the new species Bacillus altaicus are examined and their potential evolutionary and ecological causes are discussed. The authors compare their data with that in the literature and provide specific references to the work of other researchers, which is good practice. One drawback is that there are some general statements that are reminiscent of an introduction. It is recommended that more specific transitions are made between paragraphs, and that repetition of issues is avoided so as not to detract from the focus on interpreting the results. It would also be worthwhile adding a separate short paragraph outlining prospects and recommendations for further research.
Conclusion:
The identification and description of a new species, Bacillus altaicus, makes the work interesting. However, substantial revision of the methods, presentation and quality of the illustrative material is required before publication.
Author Response
We thank the reviewer for the thorough analysis of our work and valuable critical comments that substantially assisted in improving our manuscript. We are grateful for your mentioning the significance of our study to the field. We reworked the illustrative material of the study and provided a single figure (Figure 4) covering all the main findings of the research. In addition, we prepared a comprehensive supplementary material with strain-wise data. We hope that after addressing the issues mentioned, the article meets the required criteria. Please find our responses to all the comments and questions.
Introduction:
The introduction clearly demonstrates knowledge of the context of Bacillus cereus sensu lato and reveals the relevance of research into microbial diversity in Altai. However, the presentation's structure is overly detailed regarding the classification and evolution of the group, and lacks a clear formulation of the scientific problem and the novelty of this particular study. Information from different areas is intertwined in the text, and the transitions between paragraphs are sometimes inconsistent, which reduces the narrative's logic and complicates its perception. Advantages include a broad review of the literature and justification of the choice of subject, but a clear outline of the work's tasks and hypotheses is necessary to improve its structure.
Response: We agree with the reviewer that a too detailed description of the taxonomic relationships within the B. cereus s.l. group in the Introduction section is somewhat misleading and distracts from the main narrative. Moreover, it leads to a visible redundancy in the Discussion section (l. 371-381). Therefore, we transfer the passage to the Discussion to eliminate redundancy and focus on the key goal. We stated that the main aim of the study was to reveal the taxonomic position of the strain al37.1 (l. 60), which turned out to represent a novel species unit – Bacillus altaicus.
Materials and Methods
- Figure 8 does not convey any information and should be deleted.
Response: We thank the reviewer for the remark. Figure 8 has been removed from the revised manuscript as suggested.
- Isolation of bacilliform strains from soil samples is possible, but not recommended because 'bacilliform' means 'rod-shaped' and does not refer to belonging to the Bacillus genus. Scientific articles usually use 'bacilli' or 'Bacillus-type bacteria'.
Response: We appreciate the reviewer’s observation. The term bacilliform has been replaced with Bacillus-type throughout the text, as recommended (l. 548), unless the shape was only implied.
- 'Approximately 10 μl of the 547 culture was collected with the tip of a microbiological needle' is an incorrect statement, as the biomass of colonies collected from agarised media cannot be measured in volume.
Response: We agree with the reviewer’s comment. The sentence was revised to: “The culture was collected using a microbiological loop with a capacity of 10 μl,” (l. 586-587), indicating that the 10 μl refers to the loop capacity rather than the volume of colony biomass.
- Selection criteria. The section does not clearly describe the criteria for selecting strains for further analysis, and it is unclear on what basis objects were selected for detailed study from all isolates.
Response: We thank the reviewer for this observation. The section on strain selection has been expanded to clearly explain the criteria used to choose isolates for further analysis (l. 558-561).
- Representativeness and Replicates: The number of biological and technical repetitions performed for each type of experiment is not specified, which makes it difficult to assess the statistical reliability of the obtained results.
Response: We have revised the Methods to specify the numbers of biological and technical replicates for each experiment, as recommended (l.579, l. 617, l. 645, l. 771).
- Lack of control. Most of the methods described, for example those used to determine physiological properties and conduct cytotoxic testing, make no mention of the use of positive and negative control groups.
Response: Information on positive and negative controls has been added to the descriptions of the respective methods (l. 643-645, l. 762-766).
- Protocol detail. Some methods are described too briefly, providing insufficient information to reproduce individual procedures. There are also no references to generally accepted standards or detailed parameters for performing analyses, and often only the basic steps are given.
Response: The descriptions of all experimental procedures have been expanded to include detailed parameters and, where applicable, references to standard protocols.
For instance, the descriptions of sample preparation for microscopy and the reagents used are superficial. Specific concentrations, incubation times and temperatures at individual stages are not provided, making it impossible to repeat the procedure in another laboratory. The parameters for fixing and dehydrating samples for electron microscopy are not specified, nor are the imaging modes (e.g. accelerating voltage and contrast details).
Response: We have revised the microscopy section to clarify the procedures and added the missing information for electron microscopy, as suggested (l. 592-605).
Similarly, the sequences of the control and reference strains used to compare morphological features and biochemical tests are not provided — they are simply mentioned without any identifiers or information about where they came from.
Response: We appreciate the reviewer’s remark. Information on the sources of data for the reference strains is provided in the header of Table 3, which lists the main characteristics of these strains together with references to the publications from which the data were taken. In the case of sequences, we used all the available data from the RefSeq database and subsequently mined the genomes referring to the type material. We provided all sources we used in Table S1. Accession numbers of the type strains are presented in Table 1 in the main text.
- Sample storage. The storage conditions of the samples and strains under study at each stage are not fully disclosed, which may affect their condition and the results of the experiments.
Response: We thank the reviewer for this suggestion. Additional information on the storage conditions of the samples and strains has been included in the revised manuscript (l. 546-547).
- Statistical processing. The description of the statistical data processing methods used is minimal. Details of the approaches employed to analyse the significance of differences and interpret the results should be provided.
Response: We have added some detailes regarding statistical processing (l. 840-848).
- Equipment and reagents. Information about the equipment, commercial kits and reagents used is lacking; it is advisable to add details such as manufacturers, batches and specific device models.
Response: Details on the equipment, kits, and reagents (including manufacturers and device models) have been included in the corresponding sections of the Methods.
Results:
- Figure 1 is not informative and should be moved to the supplementary materials.
Response: We would kindly not agree with the reviewer on this issue. The figures here represent the morphology of the colonies. The non-rhizoidal form of the colonies is one of the major characteristics of our strain, distinguishing it from the related Bacillus mycoides species. Moreover, it is quite a common practice to provide the photographs of the colonies in the taxonomic articles, since electron microscopy alone could not provide a full phenotypic description of the bacterium. We, therefore, would like to preserve this figure in the main text as a direct and vivid illustration of the strains’ morphology.
- Microscopy quality. The quality of the microscopic images presented is extremely poor and does not meet the current publication requirements of leading international journals. In particular: The images shown in Figure 2 are blurry and have poor contrast. Additionally, the TEM images lack detail in the membranes, spores and surface structures, which are essential for differentiation within the Bacillus cereus sensu lato The scale indicators are illegible, which makes it impossible to assess the size of the cell structures objectively.
Response: We thank the reviewer for this comment. All scale bars have been corrected, and the resolution of the figures was verified to meet the journal’s requirement (≥300 dpi). Those presented in the document might be compressed by the uploading system and do not represent the original quality, which will be available in the electronic version. We also carefully reviewed descriptions of novel species within the Bacillus cereus sensu lato group and found that many publications either include micrographs of similar quality to ours or present no transmission electron microscopy images at all, focusing instead on genomic, chemotaxonomic, and phenotypic data. Below, please find notable examples with similar goals:
- doi.org/10.1099/ijsem.0.001421
- doi.org/10.1099/ijsem.0.001821
- doi.org/10.1099/ijs.0.030627-0
- doi.org/10.1099/ijsem.0.004993
- doi.10.1099/ijsem.0.006112
- doi.org/10.1016/j.syapm.2013.04.008
Illustrations 5 and 6 are unclear and uninformative. Figures 5 and 6, which illustrate metabolic potential (biosynthetic gene clusters, BGCs) and pathogenicity, are extremely poorly designed, making it difficult to interpret the results. The legends are confusing and the visual elements (colours, gradients and axis labels) are not clearly explained or correspond to the text description. The data structure remains unclear.
Figure 5 shows the distribution of BGCs, but even upon detailed examination, it is impossible to understand it: the graph is overloaded with numerous small elements, preventing a clear picture of the differences between strains from emerging.
Figure 6 shows the presence of virulence factors in the form of a heatmap, which makes it practically impossible to distinguish between clusters and compare strains.
Both figures are unsuitable for printing or electronic viewing: the elements are small, the captions overlap and the visualised values (e.g. number of genes, colour intensity) lack intuitive explanations.
Consequently, rather than helping, the illustrative material greatly hinders the perception and critical evaluation of the comparative data provided by the author.
We thank the reviewer for pointing out that the distribution of functionally important loci within distinct clades, including B. altaicus, was not displayed in a correct and readable way. Since we used a large dataset with 300 reference strains, it is barely possible to visualize all the presented data as present in the initial figures (5-6 as mentioned by the reviewer and 4 as well), we developed an appropriate trade-off. Since most of the loci were rather homogenous in terms of the clade-wise distribution, we removed Figures 4-6 from the text. Instead, we prepared a novel integrative Figure 4 showing the frequency of the loci (virulence factor, insecticidal toxins, BGCs) in the dataset. We highlighted those present in the genome of the al37.1 strain. The Figure is of sufficient printable quality. For better readability, we picked only the most frequent moieties. However, since the clade-wise analysis could be of interest to the reader, we prepared Supplementary Figures (S2-S6) with the respective data by splitting the former panels into more vivid, smaller panels. These figures represent only top hits as well. Since it could be hard to follow these figures in the Word document, we attached the scalable versions of the panel in the PDF format suitable for a detailed analysis. In addition, we provided all the identified loci with their properties, including singletons, in Supplementary Tables S5-S11 to increase the reproducibility of our findings.
Discussion
While the 'Discussion' section of the article is well structured and meets the basic requirements of microbiology journals, there are some aspects that could be improved. It opens with a brief summary of the main results, places them in the context of previous studies and focuses on novelty — an approach that is consistent with the recommendations of major scientific publishers. The interpretations of the results are detailed: the genomic, physiological and biochemical differences of the new species Bacillus altaicus are examined and their potential evolutionary and ecological causes are discussed. The authors compare their data with that in the literature and provide specific references to the work of other researchers, which is good practice. One drawback is that there are some general statements that are reminiscent of an introduction. It is recommended that more specific transitions are made between paragraphs, and that repetition of issues is avoided so as not to detract from the focus on interpreting the results. It would also be worthwhile adding a separate short paragraph outlining prospects and recommendations for further research.
We are grateful to the reviewer for mentioning these issues in the Discussion section. Since we reduced the Introduction, the taxonomy-related redundancy was removed. We improved the overall structure of the text and added more specific linkers between the individual paragraphs to ensure a coherent narrative. Moreover, as suggested, we briefly described the future prospects related to our research (l. 532-541).

Round 2
Reviewer 2 Report
Comments and Suggestions for Authors
Accept the article in its current form.